# Genome-Wide Analysis of C/S1-bZIP Subfamilies in *Populus tomentosa* and Unraveling the Role of *PtobZIP55/21* in Response to Low Energy

**DOI:** 10.3390/ijms25105163

**Published:** 2024-05-09

**Authors:** Jiangting Wu, Mengyan Zhou, Yao Cheng, Xin Chen, Shuaixu Yan, Shurong Deng

**Affiliations:** State Key Laboratory of Tree Genetics and Breeding, Key Laboratory of Silviculture of the National Forestry and Grassland Administration, Research Institute of Forestry, Chinese Academy of Forestry, Beijing 100091, China; wujiangting00@126.com (J.W.); zhoumy3639@163.com (M.Z.); cy15829011936@163.com (Y.C.); chenxincaf@caf.ac.cn (X.C.); 17837408379@163.com (S.Y.)

**Keywords:** *Populus tomentosa*, C/S1-bZIP, *bZIP55/21*, energy deprivation

## Abstract

C/S1 basic leucine zipper (bZIP) transcription factors are essential for plant survival under energy deficiency. However, studies on the responses of C/S1-bZIPs to low energy in woody plants have not yet been reported. In this study, members of C/S1-bZIP subfamilies in *Populus tomentosa* were systematically analyzed using bioinformatic approaches. Four C-bZIPs and 10 S1-bZIPs were identified, and their protein properties, phylogenetic relationships, gene structures, conserved motifs, and uORFs were systematically investigated. In yeast two-hybrid assays, direct physical interactions between C-bZIP and S1-bZIP members were observed, highlighting their potential functional synergy. Moreover, expression profile analyses revealed that low energy induced transcription levels of most *C/S1-bZIP* members, with *bZIP55* and *bZIP21* (a homolog of *bZIP55*) exhibiting particularly significant upregulation. When the expression of *bZIP55* and *bZIP21* was co-suppressed using artificial microRNA mediated gene silencing in transgenic poplars, root growth was promoted. Further analyses revealed that *bZIP55/21* negatively regulated the root development of *P. tomentosa* in response to low energy. These findings provide insights into the molecular mechanisms by which C/S1-bZIPs regulate poplar growth and development in response to energy deprivation.

## 1. Introduction

The basic leucine zipper (bZIP) family, representing one of the most diverse families of transcription factors in eukaryotes, plays a crucial role in regulating various physiological and developmental processes, as well as stress responses [1]. The bZIP domain is typically composed of 60–80 amino acids in length, containing a basic DNA binding region and a leucine zipper region [2]. The basic DNA binding region consists of approximately 20 amino acid residues, within which the conserved N-x7-R/K-x9 motif is responsible for recognizing and binding to the specific DNA sequences [2]. The leucine zipper region is mainly composed of seven repeated heptapeptide sequences, with leucine typically occupying the seventh amino acid position in each sequence [3]. However, leucine may also be replaced by other hydrophobic amino acids, i.e., isoleucine, valine, phenylalanine, and methionine [3]. The leucine zipper region typically forms an amphiphilic α-helix structure facilitating the formation of either homodimers between two identical bZIP monomers or heterodimers between two distinct bZIP monomers [2,3,4]. This diversity in dimer formation confers functional versatility to bZIP proteins. To date, the bZIP family members have been extensively identified in many plant species including *Arabidopsis thaliana* [2], *Zea mays* [4], *Oryza sativa* [5], *Solanum lycopersicum* [6], *Solanum tuberosum* [7], *Camellia sinensis* [8], *Vitis vinifera* [9], and *Malus domestica* [10].

In *A. thaliana*, the bZIP gene family is categorized into 13 groups (A-K, M, and S) [2], of which the C and S1-bZIP groups evolve from the duplication of a single algal proto-C/S ancestor [11]. It has been found that the members of C-bZIPs preferentially heterodimerize with those of S1-bZIPs to coordinate plant growth, development, and response to environmental stresses [12]. For instance, AtbZIP10 and AtbZIP25 from C-bZIPs dimerize with S1-bZIP member AtbZIP53 to activate the expression of *maturation gene* (*MAT*) during seed maturation [13]. In particular, C/S1-bZIPs play important roles in orchestrating metabolic adaptation in response to fluctuation of energy status or energy deprivation, which is typically triggered by adverse environmental conditions such as extended darkness, low light, and salt stress [14,15]. AtbZIP63 from C-bZIPs can form heterodimers with S1-bZIP members to facilitate carbohydrate and amino acid metabolism in conditions of energy starvation, leading to metabolic adaptation that enhances survival during stress [15].

In general, the transcripts of *S1-bZIP* genes feature one or more upstream open reading frames (uORFs) in their 5′ leaders [16]. These uORFs encode one or more oligopeptides, which can inhibit the translation of S1-bZIP main ORFs under conditions of high cytoplasmic sucrose levels [11]. Studies have demonstrated that the luciferase activities are significantly reduced in sucrose-treated *35S::AtbZIP11 uORF::LUC* transgenic lines compared to those without sucrose supplementation [17]. Moreover, biochemical analyses indicate that the AtbZIP11 uORF causes ribosome stalling in a sucrose-dependent manner, thus repressing the translation of the main AtbZIP11 ORF [18]. In contrast, under low carbon conditions, the translation of AtbZIP11 derepressed, coinciding with a decrease in sucrose content [19].

*Populus tomentosa*, widely cultivated across China, is notable for its rapid growth, long lifespan and high timber yield, which confers significant ecological and economic benefits [20]. The recent sequencing of the *P. tomentosa* genome provides an opportunity to characterize gene families and investigate their biological functions [21]. Given the large biomass of poplar, substantial energy and nutrients are essential for its growth. Therefore, maintaining energy homeostasis is crucial for poplar survival and adaptation to variable environmental conditions. Previous studies have reported that the C/S1-bZIP network reprogrammed metabolism under conditions of energy deficiency in herbaceous plants [12,22]. However, the C- and S1-bZIP members have yet to be comprehensively identified in poplar trees, and their responses to energy signals in woody plants remain poorly understood. In the present study, we identified four C-bZIP and 10 S1-bZIP members of *P. tomentosa*, and then analyzed their protein properties, evolutionary relationships, gene structures, conserved motif, and uORFs using bioinformatic approaches. Interactions between C and S1-bZIP members were confirmed by yeast two-hybrid assays. In addition, the expression profiles of *C/S1-bZIPs* were investigated under low energy conditions. Notably, the most responsive members, bZIP55/21, were further investigated to elucidate their roles in regulating root development under energy-deprived conditions. Our results provide valuable insights into the roles of C/S1-bZIPs in adaptation to conditions of low energy availability in poplar species.

## 2. Results

### 2.1. Genome-Wide Characterization of C/S1-bZIP Genes in P. tomentosa

There are four *C-bZIP* genes (i.e., *AtbZIP9*, *AtbZIP10*, *AtbZIP25*, and *AtbZIP63*) and five *S1-bZIP* genes (i.e., *AtbZIP2*, *AtbZIP11*, *AtbZIP44*, *AtbZIP1*, and *AtbZIP53*) in the genome of *A. thaliana* [2]. To identify the *C/S1-bZIP* genes in the genome of *P. tomentosa*, the amino acid sequences of C/S1-bZIP members in *A. thaliana* were used as query baits against the *P. tomentosa* genome. We identified four *PtoC-bZIP* genes and 10 *PtoS1-bZIP* genes, which were named according to their positions on the chromosomes (Appendix A). These 14 *PtoC/S1-bZIPs* are distributed on the nine chromosomes of *P. tomentosa* (Chr 2, 4, 5, 6, 7, 8, 9, 10, and 14). Moreover, the physical and chemical properties of PtoC/S1-bZIPs were analyzed using the software Expasy (https://www.expasy.org/, accessed on 20 December 2023) (Appendix A). The results showed that the amino acid lengths of PtoC/S1-bZIPs ranged from 139 to 497. Their molecular weights (MWs) ranged from 15.88 to 53.85 kDa, isoelectric points ranged from 5.72 to 9.66, and instability indexes ranged from 43.91 to 70.33 (Appendix A).

### 2.2. Phylogenetic Analysis of C/S1-bZIPs

To explore the evolutionary relationships and classification of C/S1-bZIPs in poplar and other plants, a phylogenetic tree of 71 C/S1-bZIP proteins from *P. tomentosa* (Pto), *A. thaliana* (At), *S. tuberosum* (St), *S. lycopersicum* (Sl), *M. domestica* (Md), and *Z. mays* (Zm) was constructed by using MEGA11 software with the Maximum Likelihood method (ML) (Appendix A). The phylogenetic tree revealed that these bZIP proteins were divided into two main clusters (Figure 1): one consisting of 25 C-bZIPs members and the other comprising 46 S1-bZIPs members. Notably, the number of *S1-bZIP* genes was particularly large in the genome of woody plants (*M. domestica* and *P. tomentosa*), where genes of this subfamily appeared in pairs (Figure 1). These results suggest that *S1-bZIPs* may have undergone gene duplication during the evolution of these woody plants.

### 2.3. Analyses of Conserved Domain, Gene Structure, and Protein Motif of C/S1-bZIP_S_

To further investigate the sequence conservation of the C/S1-bZIP protein in *P. tomentosa*, multiple sequence alignment was performed using the amino acid sequences of PtoC/S1-bZIP members (Figure 2). All C/S1-bZIP proteins of *P. tomentosa* contained the conserved bZIP domain, i.e., the basic region (N-x7-R) and the leucine zipper structure (Figure 2). The results are consistent with the findings of *A. thaliana* [2,3].

To explore the sequence structure of the C/S1-bZIP subfamilies of *P. tomentosa*, we analyzed the exon/intron structure and conserved motif composition of PtoC/S1-bZIPs (Figure 3). Gene structure analysis showed that members within the same subfamily, especially closely related ones, shared similar gene structures (Figure 3A,B). The bZIP genes of C subfamily comprised six exons, while those of S1 subfamily contained only one exon (Figure 3B). Moreover, a total of 10 conserved motifs were identified using the MEME program (Appendix A). Among them, all C/S1-bZIP proteins contained motif 1 and motif 2. However, motif 3 was only found in the PtoS1-bZIP subfamily, while motif 5, motif 6, motif 9, and motif 10 were unique to the PtoC-bZIP subfamily (Figure 3C). The presence of specific motifs in particular members suggest their potential roles in distinct biological functions.

In addition, the tertiary structure of the C/S1-bZIP proteins in *P. tomentosa* were predicted using the Phyre^2^ website (Appendix A). The results revealed that the tertiary structures of both C and S1 subfamily bZIP proteins predominantly consist of α-helices, indicating structural similarities across the subfamilies (Appendix A). Notably, the PtobZIP42 and PtobZIP25 proteins of the C subfamily exhibited significantly greater numbers of α-helices and contained different numbers of β-folds compared to other C/S1-bZIP proteins (Appendix A).

### 2.4. uORF Alignment in S1-bZIP Proteins

The S1-bZIP members typically contain the specific uORF sequences in the 5′UTR region, which can encode sucrose control peptides, playing pivotal roles in regulating the protein levels in response to energy homeostasis or stress response [23]. The uORF sequences of S1-bZIPs in *P. tomentosa*, *A. thaliana*, *S. tuberosum*, *S. lycopersicum*, *M. domestica*, and *Z. mays* were analyzed based on the conservation of uORF motifs (Figure 4). The results showed that uORFs were presented in each member of *A. thaliana*, whereas only several S1-bZIPs contained uORFs in *P. tomentosa*, *S. lycopersicum*, and *Z. mays* (Appendix A). Furthermore, multiple sequence alignment analysis of uORF-containing S1-bZIPs from these species revealed conservation of serine, leucine, isoleucine, and tyrosine within the uORFs across different species, suggesting a conserved functional role (Figure 4).

### 2.5. Interactions between C-bZIPs and S1-bZIPs

The monomers of the bZIP proteins are known to form dimers through their leucine zipper structures [2]. To explore the interactions between the members of C-bZIPs and S1-bZIPs in poplar, the full length of each *PtoC-bZIP* fused with Gal4 activation domain and the full length of each *PtoS1-bZIP* fused with GAL4 DNA-binding domain, respectively, were co-transformed into the yeast strain AH109 (Figure 5). All yeast transformants harboring the each paired PtoC-bZIP and each PtoS1-bZIP combination grew successfully on the SD/-His/-Leu/-Trp/-Ade medium and showed X-α-gal activity, confirming interactions between C- and S1-bZIP members (Figure 5). Among these, the protein–protein interaction strength between the PtobZIP42 and PtobZIP27 pair was relatively weaker compared to others. These findings demonstrate that PtoC-bZIPs physically interact with PtoS1-bZIPs, suggesting a functional network of dimerization among these proteins.

### 2.6. Expression Profiles of C/S1-bZIPs in Different Tissues and in Response to Low Energy

The expression patterns of *C/S1-bZIP* genes in various tissues of poplar were investigated (Appendix A). In the *C-bZIP* subfamily, *bZIP25* was mainly expressed in the leaves with relatively low expression in the stems and roots (Appendix A). *bZIP42*, *bZIP32*, and *bZIP7* were expressed at high levels in all tissues (Appendix A). In the *S1-bZIP* subfamily, *bZIP49* and *bZIP63* were predominantly expressed in the roots, whereas *bZIP44* and *bZIP27* showed high expressions in both stems and roots (Appendix A). The transcript levels of *bZIP75*, *bZIP15*, *bZIP21*, *bZIP55*, *bZIP33*, and *bZIP6* were very high in the leaves, stems and roots (Appendix A).

Furthermore, we analyzed the expression profiles of *C/S1-bZIPs* in response to low energy conditions at different times (extended dark for 24 h, 48 h, and 96 h) in the poplar roots (Figure 6). In the *C-bZIP* subfamily, the expression levels of all four *C-bZIP* genes were significantly upregulated under different dark conditions (Figure 6). Notably, *bZIP25* showed the highest upregulation among the *C-bZIP* subfamily, peaking at 48 h, with an 11-fold higher level than that of the control (Figure 6). Similarly, all *S1-bZIPs*, except for *bZIP75*, *bZIP33*, and *bZIP6*, showed marked increases in transcript levels under dark conditions (Figure 6). Among them, *bZIP55* was the most strongly inducted by darkness (Figure 6). The relative expression of *bZIP55* reached the highest level in the roots exposed to darkness at 48 h, with a 23-fold higher level compared to the control (Figure 6). Moreover, *bZIP21*, *bZIP27*, and *bZIP44* also exhibited relatively high expression levels under dark treatments (Figure 6). These results suggest that poplar C/S1-bZIPs probably play important roles in response to low energy.

### 2.7. PtobZIP55/21 Inhibited Root Development in Response to Low Energy

To elucidate the roles of C/S1-bZIPs in the root development of poplar under low energy, the paralogous pairs *bZIP55/21* were selected as candidates for further analyses according to the expression profiles of *C/S1-bZIPs*. As *PtobZIP55* and *PtobZIP21* are homologous genes with high sequence similarity, and they have similar tissue-specific expression patterns and responses to darkness, the two genes may function redundantly. Thus, amiRNA technology was used for silencing *bZIP55* and *bZIP21* simultaneously (Appendix A). Seven transgenic *bZP55/21*-amiRNA lines of *P. tomentosa* were generated and validated using genomic PCR (Appendix A). Lines amiRNA-3 and amiRNA-6 with the lowest expression levels of *bZIP55* and *bZIP21* were selected for further analyses (Appendix A).

To assess the impact of *bZIP55/21* on root development under low energy stress, the *bZIP55/21*-amiRNA and WT poplars were treated with control light or low light conditions for four weeks (Figure 7A). Under control light, the lengths of adventitious root (AR), lateral root (LR), and LR number were markedly increased in the *bZIP55/21*-amiRNA lines compared to those of WT plants (Figure 7B–D). Compared with control light, low light induced reductions in the AR length, LR length, and LR number of *bZIP55/21*-amiRNA and WT poplars (Figure 7B–D). However, under low light conditions, the lengths of AR and LR and the number of LR were significantly increased in the *bZIP55/21*-amiRNA lines in comparison with WT (Figure 7B–D). These results suggest that bZIP55/21 negatively regulate the root development of *P. tomentosa* in response to low energy.

## 3. Discussion

The bZIP gene family, one of the largest families in eukaryotes, includes C- and S1-bZIPs, which originated as sister groups prior to the evolution of land plants [1,11]. Members of these subfamilies can form specific heterodimers to regulate plant growth and development in response to environmental signals [24]. Our study focused on the identification and analyses of C/S1-bZIP members in *P. tomentosa*. We also investigated the heterodimerization interactions between C/S1-bZIPs and their roles in modulating root growth under low energy conditions.

In this study, four *C-bZIPs* and 10 *S1-bZIPs* were identified in the *P. tomentosa* genome. The number of C-bZIP members in *P. tomentosa* is comparable to *A. thaliana* (four members) [2] and *S. lycopersicum* (three members) [6] and is larger than *S. tuberosum* (one member) [7], but less than *M. domestica* (six members) [10] and *Z. mays* (seven members) [4], respectively. On the other hand, the number of *S1-bZIPs* in *P. tomentosa* is double that found in *A. thaliana* (five members) [2] and *Z. mays* (five members) [4], slightly higher than *S. lycopersicum* (seven members) [6], and similar to *M. domestica* (nine members) [10] and *S. tuberosum* (ten members) [7]. The presence of multiple paralogous pairs of *C/S1-bZIPs* in *P. tomentosa*, such as *PtobZIP32/7*, *PtobZIP49/63*, *PtobZIP75/15*, *PtobZIP44/27*, *PtobZIP55/21*, and *PtobZIP33/6*, suggests the occurrence of genome duplication events. The ancestral genome of poplar is known to have experienced at least three genome-wide duplications, one coinciding with the divergence from the *Arabidopsis* lineage [25]. Typically, in *Arabidopsis*, one gene corresponds to a pair of homologous loci in the genome of poplar [25]. Interestingly, PtoC/S1-bZIPs are initially clustered closely with AtC/S1-bZIPs compared to other species, indicating a closer evolutionary relationship. Multiple sequence analyses verified that all C/S1-bZIP members of *P. tomentosa* possess the characteristic N-X7-R motif and the leucine zipper structure across, consistent with the universal features of bZIP proteins observed in other plants such as *A. thaliana* [2]. These findings suggest that PtoC/S1bZIPs are evolutionarily conserved.

The genomic structures of genes provide insights into the origin and evolution of specific genes and families [26]. In our analysis, *PtoC-bZIPs* in *P. tomentosa* typically contained five introns, while *PtoS1-bZIPs* lacked introns entirely. These gene structure characteristics correspond to observations of *C/S1-bZIPs* in the monocotyledons (i.e., *Z. mays* and *Oryza sativa*) [4,5] and dicotyledons (i.e., *A. thaliana*, *S. lycopersicum*, and *M. domestica*) [2,6,10], further indicating a conservation in the evolution of *PtoC/S1-bZIP* subfamilies. Moreover, the clustered PtoC/S1-bZIP members share similar conserved motifs, whereas those from different branches exhibit distinct motifs, indicating functional differentiation among the PtoC/S1-bZIPs in different branches.

Previous studies have demonstrated that the S1-bZIPs expression can be downregulated by sucrose at the post-transcription level with high sucrose concentration inhibiting their translation and consequently reducing protein abundance [17]. This regulatory mechanism, which controls S1-bZIPs levels according to cellular energy status, is termed “sucrose-induced repression of translation” (SIRT) and is associated with the translation of the uORF leader sequences in *S1-bZIPs* transcripts [27,28]. In the present study, uORFs were present in the 5′UTR regions of all five S1-bZIPs in *A. thaliana*. In contrast, uORFs were identified in 3 out of 7 S1-bZIPs in *S. lycopersicum*, 2 out of 5 in *Z. mays*, and 5 out of 10 in *P*. *tomentosa*, whereas none were detected in S1-bZIPs of *M. domestica* and *S. tuberosum*. Our results suggest that the variations in uORF occurrences among different species potentially lead to distinct regulatory effects on S1-bZIPs translation and thus affect their functions in response to environmental cues. Amino acid mutational analysis further revealed that the serine, leucine, and tyrosine in the uORF region of AtbZIP11 are essential for SIRT [29]. Interestingly, our results indicate that these three amino acids are highly conserved in the uORFs of S1-bZIPs in *A. thaliana*, *S. lycopersicum*, *Z. mays*, and *P. tomentosa*.

Through leucine zipper domains, bZIP monomers typically bind to DNA sequences in the form of dimers, which provides significant combinatorial flexibility in transcriptional regulation [30,31,32]. In *A. thaliana*, yeast two-hybrid (Y2H) assays have demonstrated that C-bZIPs heterodimerized with S1-bZIPs [33]. However, homodimerization between their orthologous members was rarely observed [33]. In line with the Y2H results, protoplast two-hybrid systems revealed strong interaction affinities between the C-bZIPs and S1-bZIPs, whereas the homodimers formation within the same subfamily were extremely weak [33]. Similar patterns were also found in *M. domestica*, where the MdC-bZIP members showed strong preference for heterodimerization with MdS1-bZIP members [34]. Consistently, our findings indicate that each C-bZIP member interacts with S1-bZIPs, yet the strength of these interactions may vary among different pairs.

The C/S1-bZIP network has been established as a signaling hub for coordinating plant development and stress response [12]. The expression of *AtC/S1-bZIP* genes throughout Arabidopsis plants, with partially overlapping profiles, accounts for the functional redundancy observed and the capacity for C/S1-bZIP heterodimer formation [27,35]. Snf1-RELATED PROTEIN KINASE 1 (SnRK1), a central metabolic kinase, plays a pivotal role in responding to low energy signal. Transcriptomic and molecular analyses have identified S1-bZIPs as SnRK1-dependent regulators under energy deprivation [15]. Notably, the results of protoplast two-hybrid assays revealed a significant increase in the interaction between bZIP2 (S1-bZIP) and bZIP63 (C-bZIP) when co-expressed with SnRk1 [15]. Furthermore, SnRK1 has been shown to phosphorylate bZIP63 of C-bZIPs in *A. thaliana*, enhancing its heterodimerization with S1-bZIP members and thereby regulating metabolic reprogramming to maintain energy homeostasis under starvation conditions [36]. AtbZIP1 and AtbZIP53 reprogram carbohydrate and amino acid metabolism to serve the energy demand under salt stress [14]. In *M. domestica*, the MdC/S1-bZIP network has been found to negatively regulate drought tolerance and low energy-induced senescence in a functionally redundant manner [34]. In the present study, most members of *PtoC/S1-bZIP* subfamilies showed partially overlapping expression patterns, with high expression in the roots and significant upregulation under dark conditions, suggesting that poplar C/S1-bZIP members likely play crucial and redundant roles in response to low energy.

Plants often encounter various stresses in nature, such as pest infestations and environmental fluctuations, which disrupt energy homeostasis, leading to impaired growth and development [12,37,38]. Maintaining energy homeostasis is crucial for plant survival. In *A. thalinana*, C/S1-bZIPs regulate carbohydrate and amino acid metabolism by binding directly to the protomers of *PROLINE DEHYDROGENASE* (*ProDH*) and *ASPARAGINE SYNTHETASE1* (*ASN1*) under energy limitation [39,40]. Similarly, overexpression of *SlbZIP1* in *S. lycopersicum* and *tbz17* in *Nicotiana tabacum* also activate *ProDH* and *ASN1* gene expression [29,41]. The expression levels of close homologs *AtbZIP2*, *AtbZIP11*, and *AtbZIP44* were suppressed by sucrose and induced by low energy, highlighting their roles in energy responses [17]. Recent studies have shown that the primary root growth under starvation is significantly less reduced in the *AtbZIP2/11/44*-amiRNA lines compared to WT [42]. Furthermore, molecular analyses suggested *AtbZIP2/11/44* inhibit root growth by activating the transcriptional repressor *INDOLE-3-ACETIC ACID PROTEIN 3* (*AtIAA3*) under low energy conditions [42]. In *P. tomentosa*, the paralogous genes *PtobZIP55* and *PtobZIP21*, sharing close homology with *AtbZIP2*/*11*/*44*, were the most upregulated *C/S1-bZIP* genes under low energy conditions, suggesting similar biological functions to those of AtbZIP2/11/44. Silencing of *PtobZIP55*/*21* by amiRNA technology increased the AR length, LR length, and LR number in the transgenic poplars under low energy conditions, further indicating their negative impacts on root growth and development. Consistent with our findings, overexpression of *PdebZIP53* (a poplar homolog of *AtbZIP53*) inhibited the AR development by activating the expression of *PdebIAA4* [43]. However, overexpression of *PtabZIP1*-like (homologous to *AtbZIP1*) promoted lateral root primordia initiation and LR development [44]. Our phylogenetic analysis identified PtobZIP49 and PtobZIP63 as the orthologs of AtbZIP1, which were located on a distant branch from PtobZIP55/21, indicating divergent functional roles among S1-bZIP family members across separate evolutionary branches. Under low energy stress conditions, such as low light exposure, plants tend to slow down or halt growth in order to conserve energy and resources. We showed that low-energy-induced inhibition in root development was ameliorated in the *PtobZIP55*/*21*-amiRNA lines compared to WT, suggesting that the reduction of *PtobZIP55*/*21* transcripts diminishes their response to low light stress in the amiRNA poplars. Taken together, these results indicate that PtobZIP55/21 play pivotal roles in regulating root development in response to low energy.

## 4. Materials and Methods

### 4.1. Identification of C/S1-bZIP Subfamilies in Populus tomentosa

To identify the members of C/S1-bZIP subfamilies in *P. tomentosa,* the homologous C/S1-bZIP protein sequences in *A. thaliana* were used as query sequences in the BLASTP program to search for C/S1-bZIP proteins in the genome database of *P. tomentosa* (https://www.ncbi.nlm.nih.gov/datasets/genome/GCA_018804465.1/, accessed on 20 December 2023). The obtained C/S1-bZIP proteins were then examined using the programs of HMMER (https://www.ebi.ac.uk/Tools/hmmer/search/hmmsearch/, accessed on 20 December 2023), PFAM (http://pfam.xfam.org/, accessed on 20 December 2023), and SMART (http://smart.embl-heidelberg.de/, accessed on 20 December 2023).The chromosomal location, number of amino acid, molecular weights (MWs), isoelectric points (PI), and instability index of C/S1-bZIP subfamilies in *P. tomentosa* were predicted by ExPASy (https://web.expasy.org/protparam/, accessed on 20 December 2023).

### 4.2. Phylogenetic, Multiple Sequence Alignment, Gene Structure, Conserved Motif, and uORF Analyses of C/S1-bZIPs in P. tomentosa

The C/S1-bZIPs sequences of *P. tomentosa* (Pto), *A. thaliana* (At), *S. tuberosum* (St), *S. lycopersicum* (Sl), *M. domestica* (Md), and *Z. mays* (Zm) were downloaded from the Phytozome and NCBI databases. The phylogenetic tree was constructed using MEGA11 by the Maximum Likelihood method (bootstrap analysis for 1000 replicates) [45]. ClustalX (http://www.clustal.org/clustal2/, accessed on 20 December 2023) was used for the multi-sequence alignment of the PtoC/S1-bZIP subfamilies of *P. tomentosa*. To analyze the gene structures of the *PtoC/S1-bZIP* subfamilies, the genomic sequences and corresponding coding sequences of *PtoC/S1-bZIPs* were downloaded from the genome database of *P. tomentosa*. Then the gene structure diagrams were drawn by the gene structure visualization server GSDS (https://gsds.gao-lab.org/index.php/, accessed on 20 December, 2023). The protein sequences of PtoC/S1-bZIPs were submitted to the MEME website (https://meme-suite.org/meme/index.html/, accessed on 20 December 2023) to predict the conserved motifs with default parameters (the maximum number of motifs set to 10). The tertiary structures of the PtoC/S1-bZIP proteins were predicted on the Phyre^2^ website (http://www.sbg.bio.ic.ac.uk/phyre2/html/, accessed on 20 December 2023). The uORFs of S1-bZIPs were identified using Open Reading Frame Finder (https://www.ncbi.nlm.nih.gov/orffinder/, accessed on 20 December 2023).

### 4.3. Plant Cultivation and Treatments

The plantlets of *P. tomentosa* were maintained through micropropagation on a woody plant medium (WPM) in a climate chamber (day/night temperature, 25 °C/18 °C; relative air humidity, 50–60%; light per day, 16 h). For treatment with low energy, the populus plantlets in the climate chamber were transferred to a temperature-light gradient incubator maintained dark conditions for 24 h, 48 h, and 96 h, respectively.

### 4.4. Yeast Two-Hybrid (Y2H) Assay

The Y2H assays were carried out according to the method [46] with minor modifications. Briefly, the full-length coding sequences (CDSs) of *PtoS1-bZIPs* were cloned and inserted into the pGBKT7 vector via the EcoR I and Sal I sites, respectively, to create PtoS1-bZIPs-BD recombinant constructs. The CDSs of *PtoC-bZIPs* were cloned and inserted into the pGADT7 vector via the EcoR I and Sac I sites, respectively, to create PtoC-bZIPs-AD recombinant constructs. The plasmids of BD and AD recombinant constructs were co-transformed into the yeast strain AH109. The yeast cells were cultured on SD/-Leu/-Trp solid medium for four days, and positive yeast clones were verified by PCR. Then the positive yeast strains were grown in an SD/-Leu/-Trp liquid medium and spotted on either an SD/-Leu/-Trp solid medium or an SD/-His/-Leu/-Trp/-Ade solid medium supplemented with X-α-gal. These plates with yeast cells were incubated at 28 °C for four days and then photographed.

### 4.5. Expression Patterns of C/S1-bZIP Genes

To explore the expression patterns of *C/S1-bZIPs* in various tissues of poplar, the high-throughput data (GSE81077) was searched in the NCBI-GEO database (https://www.ncbi.nlm.nih.gov/geo/query/acc.cgi/, accessed on 20 December 2023), the data was then quantified using the Transcripts Per Kilobase of exon model per Million mapped reads (TPM) algorithm, and the expression heatmap was produced by TB tools [47].

### 4.6. RNA Extraction and Quantitative RT-PCR

To explore the expression profiles of *C/S1-bZIPs* in response to low energy, the total RNA from the roots of *P. tomentosa* was extracted using the CTAB method with minor modifications [48]. The cDNA was synthesized using PrimeScript™ RT reagent Kit (RR047A, Takara, Dalian, China). The quantitative RT-PCR was performed on a real time system (LightCycler^®^ 480 II, RoChe, Rotkreuz, Switzerland). The expression levels of PtoC/S1-bZIPs were calculated using the 2^−ΔΔCT^ method [49] with poplar *Ubiquitin* (*UBQ*) gene as internal reference. The primers were listed in Appendix A.

### 4.7. Plasmid Construction and Plant Transformation

To generate amiRNA construct, the target site was designed for the conserved sequences between PtobZIP55 and PtobZIP21 using the WMD3 web microRNA designer (http://wmd3.weigelworld.org/cgibin/webapp.cgi/, accessed on 28 December 2023) as described by Aung et al. [50]. One of the recommended amiRNAs (CTCGTTCAGATCTTACTCTTT) was selected and cloned into the pGWB2 vector using pENTR™/D-TOPO^®^ cloning kit (K240020SP, Invitrogen, Paisley, UK) and Gateway^®^ LR Clonase^®^ II Enzyme mix kit (11789-020, Invitrogen, Paisley, UK). The amiRNA construct was transformed into Agrobacterium tumefaciens strain GV3101 and confirmed by PCR. The transgenic plants of *P. tomentosa* were produced as described previously [51]. Putative transgenic plants were selected on WPM medium supplemented with 9 mg L^−1^ hygromycin. The genomic DNA of putative transgenic lines were extracted using the CTAB method [52] and identified by PCR for successful transformation of the plasmids. To further identify the *PtobZIP55/21*-amiRNA transgenic lines, RT-qPCR was used to screen the lines with the lowest expression levels of *PtobZIP55* and *PtobZIP21*. For analyses of root development, the selected *PtobZIP55*/*21*-amiRNA transgenics and WT plants were cultivated in WPM solid medium under a either control light (150 μmol m^−2^ s^−1^) or low light conditions (40 μmol m^−2^ s^−1^), and the root growths were then observed after four weeks.

### 4.8. Statistical Analysis

Statistical analysis was carried out using Statgraphics Centurion XVI.I (STN, St. Louis, MO, USA). One-way ANOVA was used to the analyses of the expression profiles of PtoC/S1-bZIPs in response to low energy and the identification of *PtobZIP55*/*21*-amiRNA transgenic lines. Two-way ANOVA tests were used to analyze the influences of genotypes (WT and transgenics) and low energy treatments on root development. The data were checked for normality before statistical analyses. According to the analysis of ANOVA F-test, the differences between the means were considered to be significant when the *p*-value was less than 0.05. Posteriori comparisons of the means were performed according to the least significant difference method.

## 5. Conclusions

In the present study, four C-bZIPs and 10 S1-bZIPs were identified in *P. tomentosa*. Bioinformatic analyses indicated that PtoC/S1-bZIP subfamilies were conserved during the evolution. Additionally, the specific uORF sequences were identified in the 5′UTR region of some *PtoS1-bZIPs* members. The results of Y2H assays revealed that each C-bZIP member interacted with S1-bZIP members. Notably, most *C/S1-bZIPs* exhibited increased expression levels under low energy conditions, with *bZIP55* and its homolog *bZIP21* showing significant upregulation. Further genetic analyses suggested that bZIP55/21 play a role in inhibiting root growth and development in response to low energy. These findings provide a foundation for further investigation of the biological functions and molecular mechanisms of C/S1-bZIPs in response to environmental stress in poplars.

## Figures and Tables

**Figure 1 ijms-25-05163-f001:**
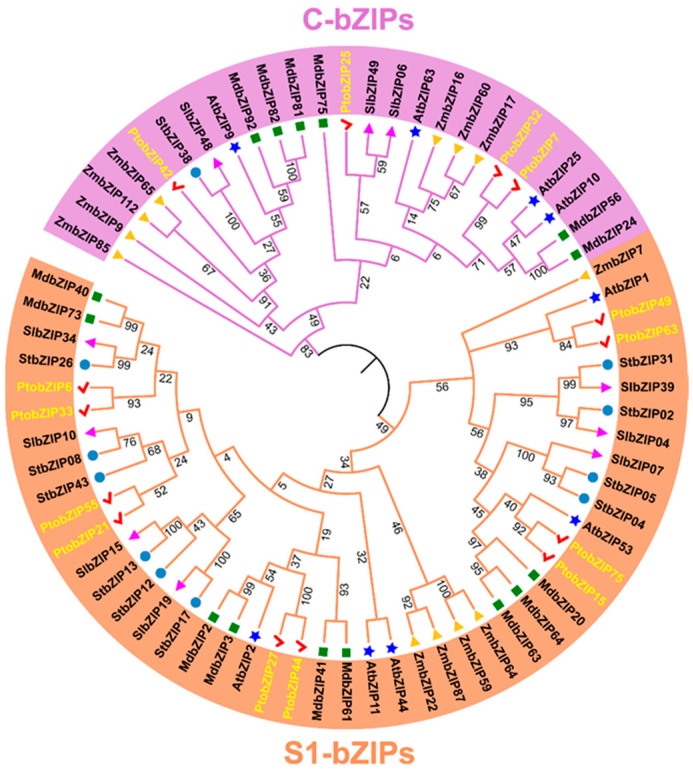
Phylogenetic analysis of C and S1-bZIP subfamilies in *P. tomentosa* (Pto), *A. thaliana* (At), *M. domestica* (Md), *S. lycopersicum* (Sl), *S. tuberosum* (St), and *Z. mays* (Zm). Purple and orange branches represent C-bZIPs and S1-bZIPs, respectively. Accession numbers of *C/S1-bZIP* genes are listed in Appendix A.

**Figure 2 ijms-25-05163-f002:**
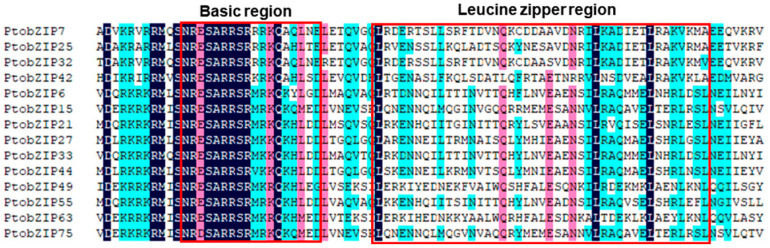
Comparison of amino acid sequences of C/S1-bZIP members in *P. tomentosa*. The conserved basic region and leucine zipper region are marked with red boxes.

**Figure 3 ijms-25-05163-f003:**
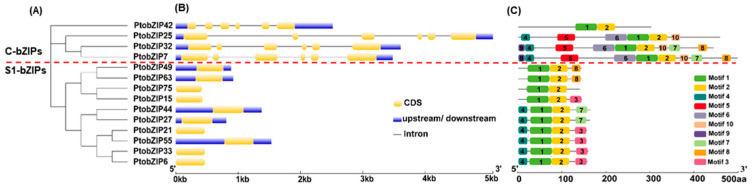
Schematic diagram of conserved motifs and gene structure of C/S1-bZIP subfamilies in *P. tomentosa*. (**A**) The phylogenetic tree of C/S1-bZIP proteins based on amino acid sequences. (**B**) The exon/intron structure of the *C/S1-bZIP* genes. UTRs, exons, and introns are shown in blue boxes, yellow boxes, and grey lines, respectively. (**C**) The conserved motifs of C/S1-bZIP proteins were identified by MEME. Different colored boxes represent the different motifs of proteins. The scales at the bottom in panel (**B**,**C**) indicate the sequence lengths. The conserved motifs are listed in Appendix A.

**Figure 4 ijms-25-05163-f004:**
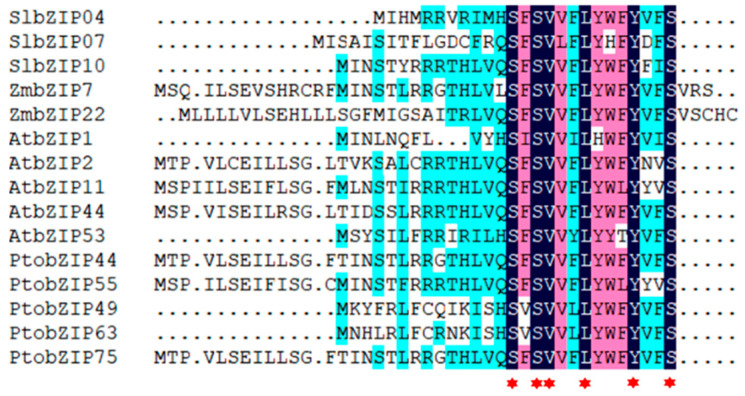
Alignment of S1-bZIP uORFs in *S. lycopersicum* (Sl), *Z. mays* (Zm), *A. thaliana* (At), and *P. tomentosa* (Pto). Conserved amino acids within uORFs are represented by red asterisks.

**Figure 5 ijms-25-05163-f005:**
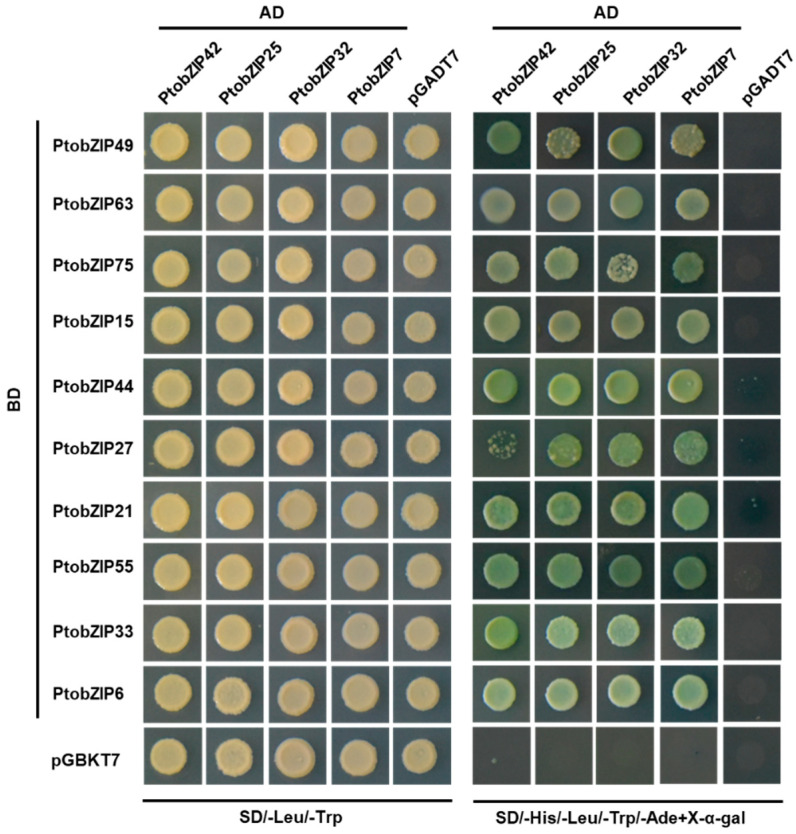
PtoS1-bZIPs interact with PtoC-bZIPs by the Y2H assays. The CDSs of 10 *PtoS1-bZIP* genes were inserted into pGBKT7 vector, respectively. The CDSs of four *PtoC-bZIP* genes were inserted into pGADT7 vector, respectively. The PtoS1-bZIPs-BD and PtoC-bZIPs-AD constructs were co-transformed into yeast AH109 cells. Yeast cells were plated on either SD/-Leu/-Trp or SD/-His/-Leu/-Trp/-Ade containing X-α-gal.

**Figure 6 ijms-25-05163-f006:**
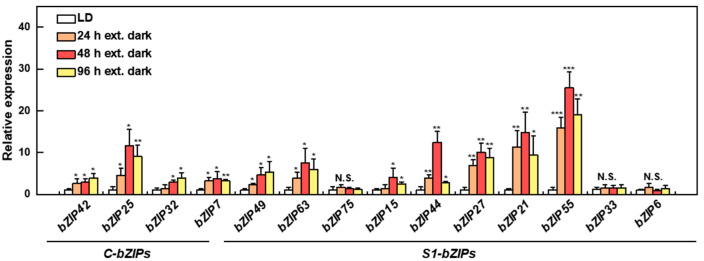
The expression patterns of *C/S1-bZIP* genes in response to low energy at different times in the roots of *P. tomentosa*. The expression level of each gene was set to 1 under long day (LD) conditions, and the corresponding fold change in expression level of each gene was calculated under extended dark (24 h, 48 h, and 96 h) conditions, respectively. The bars indicate means ± SE (n = 4). Asterisks on the bars indicate significant differences in the relative expression levels of *C/S1-bZIPs* between the treatments. *: *p* < 0.05; **: *p* < 0.01; ***: *p* < 0.001. N.S.: no significant difference between control and dark treatments.

**Figure 7 ijms-25-05163-f007:**
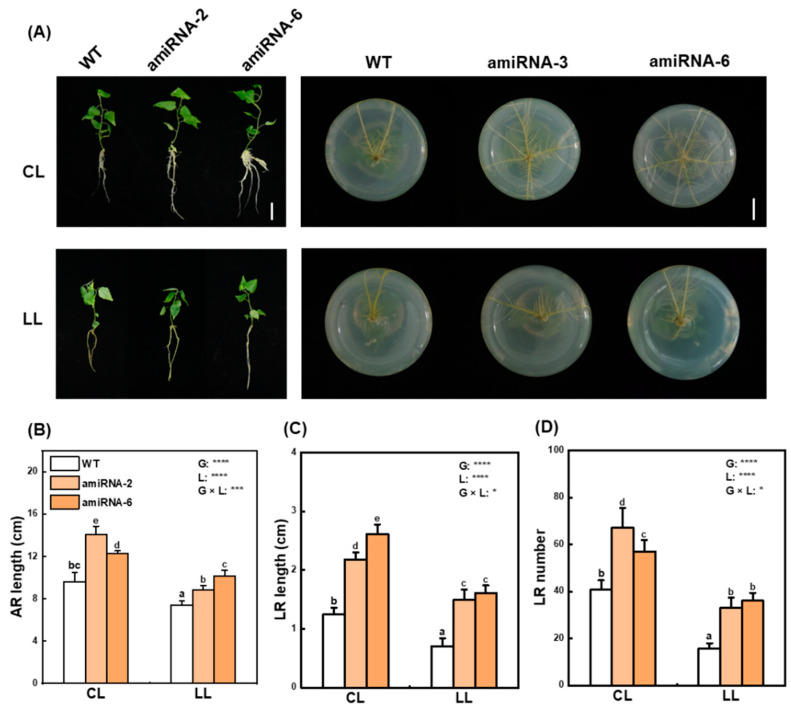
Growth performance of WT and transgenic lines in response to low energy. (**A**) Root development of WT, *PtobZIP55/21*-amiRNA-2 and *PtobZIP55/21*-amiRNA-6 under either control light (CL) or low light (LL) conditions for four weeks. Bar = 3 cm. (**B**–**D**) The average AR lengths (**B**), LR lengths (**C**), and LR numbers (**D**) of WT and transgenic lines under either CL or LL conditions. The bars indicate means ± SE (n = 4). Bars labeled with different letters (a, b, bc, c, d, e) showed significant differences at *p* < 0.05. *p*-values of the two-way ANOVAs of genotype (G), Light (L), and their interactions (G × L) are also indicated. *: *p* < 0.05; ***: *p* < 0.001; ****: *p* < 0.0001.

## Data Availability

Data are contained within the article and Appendix A.

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
