# Peer review of "Genome-Wide Analysis of C/S1-bZIP Subfamilies in Populus tomentosa and Unraveling the Role of PtobZIP55/21 in Response to Low Energy"

_ijms, 2024, doi:10.3390/ijms25105163_

Round 1
Reviewer 1 Report
Comments and Suggestions for Authors
The manuscript Genome-wide analysis of C/S1-bZIP subfamilies in Populus to mentosa and unraveling the role of PtobZIP55/21 in response to low energy is very nicely composed and work itself is very interesting and can be very useful for the readers of the journal, quality of the manuscript is high and deserve for publication, however I suggest following changes
Language: Extensive language correction needed throughout
Abstract is too long, i suggest author should delete following sentances from the abstract
"Maintaining energy homeostasis is crucial for plant survival under low energy stress. The 13 members of C and S1subfamilies of basic leucine zippers (bZIPs) play pivotal roles in managing the 14 low energy response in plants. However, the comprehensive characterization of C/S1-bZIP mem- 15 bers in Poplar and their functions during low energy stress have yet to be defined."
Introduction is well arranged however author should update the citations
Experimental is very nicely presented and needs no change
Results and Discussion: in this section authors have the amino acid sequences of C/S1-bZIP members in A. thaliana were used as query baits against the genome of P. tomentosa. Four PtoC-bZIP genes and ten PtoS1-bZIP genes were identified and named according to their positions on the chromosomes, results seems quite interesting. Author carefully check the typo and syntax erros in the result and discussion section
Conclusion: I suggest authors to re write the conclusion and to make it more clear and attractive
Decision: Minor revision suggested
Comments on the Quality of English LanguageExtensive editing of English language required
Author Response
The manuscript Genome-wide analysis of C/S1-bZIP subfamilies in Populus tomentosa and unraveling the role of PtobZIP55/21 in response to low energy is very nicely composed and work itself is very interesting and can be very useful for the readers of the journal, quality of the manuscript is high and deserve for publication, however I suggest following changes
General answer: Thank you for reviewing our manuscript. Your positive comments and suggestions are highly appreciated. Now we’ve made a revision according to your suggestions. Specific changes have been listed in the following section.
1 Language: Extensive language correction needed throughout
Answer: Thank you for spending your precious time on reviewing our manuscript. Now the manuscript has been carefully checked, and the typos and grammar errors have been corrected. We hope the revised manuscript could be acceptable for you.
2 Abstract is too long, i suggest author should delete following sentences from the abstract “Maintaining energy homeostasis is crucial for plant survival under low energy stress. The members of C and S1subfamilies of basic leucine zippers (bZIPs) play pivotal roles in managing the low energy response in plants. However, the comprehensive characterization of C/S1-bZIP members in Poplar and their functions during low energy stress have yet to be defined.”
Answer: Thanks for your professional suggestion. We have deleted these sentences from the abstract, and added concise sentences “C/S1 basic leucine zipper (bZIP) transcription factors are essential for plant survival under energy deficiency. However, studies on the response of C/S1-bZIPs to low energy in woody plants have not yet been reported.” to introduce the importance of our study.
3 Introduction is well arranged however author should update the citations.
Answer: Thanks for your thoughtful suggestion. Now we have update recent references in the introduction.
4 Experimental is very nicely presented and needs no change.
Answer: Thanks for your careful reading. We appreciate your positive comments.
5 Results and Discussion: in this section authors have the amino acid sequences of C/S1-bZIP members in A. thaliana were used as query baits against the genome of P. tomentosa. Four PtoC-bZIP genes and ten PtoS1-bZIP genes were identified and named according to their positions on the chromosomes, results seem quite interesting. Author carefully check the typo and syntax errors in the result and discussion sections.
Answer: Thanks for your professional suggestion. We have checked the typo and syntax errors and made careful modifications in the result and discussion sections. Thank you again for your positive comments and valuable suggestions to improve the quality of our manuscript.
6 Conclusion: I suggest authors to rewrite the conclusion and to make it more clear and attractive.
Answer: Thanks for your suggestion. To clarify and attract the conclusion, we have changed the sentence in the previous version “In the present study, four C-bZIPs and ten S1-bZIPs were identified in P. tomentosa. The characteristics of C/S1-bZIP subfamily members, including phylogenetic relationships, multiple sequence alignments, gene structures, conserved motifs, and uORF identification, were analyzed using bioinformatic approaches. Additionally, Y2H assays revealed that each C-bZIP member can interact with S1-bZIP members. The expression profile analysis of C/S1-bZIPs showed that most members of C/S1-bZIP subfamilies were highly expressed in the roots and their expression was induced by low energy. Notably, bZIP55 and its homologous gene bZIP21 exhibited the highest expression levels among C/S1-bZIP members under low energy conditions. Further genetic analyses revealed a critical role of bZIP55/21 in responding to the low energy by inhibiting root growth and development. These results provide a foundation for further investigation of the biological functions of C/S1-bZIP genes in poplars and their molecular mechanisms in response to low energy.” to “In the present study, four C-bZIPs and ten S1-bZIPs were identified in P. tomentosa. Bioinformatic analyses indicated that PtoC/S1-bZIP subfamilies were conserved during the evolution. Additionally, the specific uORF sequences were identified in the 5’UTR region of some PtoS1-bZIPs members. The results of Y2H assays revealed that each C-bZIP member interacted with S1-bZIP members. Notably, most C/S1-bZIPs exhibited increased expression levels under low energy conditions, with bZIP55 and its homolog bZIP21 showing significant upregulation. Further genetic analyses suggested that bZIP55/21 play a role in inhibiting root growth and development in response to low energy. These findings provide a foundation for further investigation of the biological functions and molecular mechanisms of C/S1-bZIPs in response to environmental stress in poplars.” in this revision on page 11 and page 12 lines 455-464.

Reviewer 2 Report
Comments and Suggestions for Authors
§ It is well-directed work of Genome-wide analysis of C/S1-bZIP subfamilies in Populus to mentosa and unraveling the role of PtobZIP55/21 in response to low energy
§ and falls wihtin the scope of the Journal. Although the manuscript is written and explained clearly but some major amendments are required brefore it going to further proceedings.
§ "exihibiting" should be corrected to "exhibiting" in line 23.
§ The terminology used to describe gene names and functions needs to be consistent throughout the manuscript. The switch between "bZIP55 and its homologous gene bZIP21" could be confusing. It's crucial to clarify whether bZIP21 is a homolog, isoform, or distinct gene with similar functions to ensure clear communication.
§ The description of bioinformatic approaches and the yeast two-hybrid assays could be insufficiently detailed for reproducibility. Ensuring that enough details are provided about the methods and tools used in bioinformatics analysis and laboratory experiments is essential for scientific validity.
§ The term "artificial microRNA mediated gene silencing" is mentioned, but there's no detail on the specific constructs or the efficiency of the silencing, which are crucial for assessing the reliability of the experimental outcomes.
§ "active the expression of seed maturation gene" in line 55 should be corrected to "activate the expression of seed maturation genes."
§ The manuscript could benefit from consistent and clear definitions, especially when discussing complex mechanisms. For instance, the role of uORFs in regulating translation under different sucrose conditions could be expanded for clarity and depth.
§ In line 57, "This uORF encodes a sucrose control peptide," could specify that it's not just "a" but specifically the uORFs discussed that encode this peptide, enhancing clarity.
§ The use of terms like "homodimers or heterodimers" in lines 42-43 should be explained briefly to ensure clarity for readers not familiar with the terminology.
§ Reference are too old in the introduction and discussion section, In so doing, it is suggested that the following articles be used as a reference.1.Pyrosequencing uncovers a shift in bacterial communities across life stages of Octodonta nipae (Coleoptera: Chrysomelidae)." 2. The endosymbiotic Wolbachia and host COI gene enable to distinguish between two invasive palm pests; coconut leaf beetle, Brontispa longissima and hispid leaf beetle, Octodonta nipae. Journal of Economic Entomology. Furthermore, The authors should try to provide some different keywords. This would increase the visibility of the paper by search engines if accepted for publication by the journal.
§ Conclusion section looks like the introduction, it must concise and only provide conclusion of study resutls and give some future direction, how your study can provide basic data for resercher working on microbial role.
Scientific questions
§ How do bZIP55 and bZIP21 specifically contribute to the regulation of root growth under low energy conditions? What are the downstream targets or pathways mediated by these genes during stress response?
§ What are the molecular mechanisms underlying the interactions between C-bZIPs and S1-bZIPs? How do these interactions influence the transcriptional regulation under low energy conditions?
§ Is there functional redundancy among the C/S1-bZIP members, and if so, how does this redundancy contribute to the robustness of the low energy stress response in Populus tomentosa?
§ How do the functions of C/S1-bZIP members in Populus tomentosa compare with those in other plant species, particularly in terms of evolutionary adaptations to energy stress?
§ Beyond root growth inhibition, how does the suppression of bZIP55 and bZIP21 affect other physiological and developmental processes in Populus tomentosa under low energy conditions?
§ What are the upstream signals and factors that regulate the expression of bZIP55 and bZIP21 during low energy stress?
Comments on the Quality of English Languageminor editing is required
Author Response
It is well-directed work of Genome-wide analysis of C/S1-bZIP subfamilies in Populus tomentosa and unraveling the role of PtobZIP55/21 in response to low energy and falls within the scope of the Journal. Although the manuscript is written and explained clearly but some major amendments are required before it going to further proceedings.
General answer: Thank you for spending your precious time on reviewing our manuscript. Your recognition and comments on our manuscript are highly appreciated. Now we have made a revision according to your suggestions.
1 “exihibiting” should be corrected to “exhibiting” in line 23.
Answer: Thanks for your careful reading. Now we have corrected “exihibiting” to “exhibiting” in this revision on page 1 line 20.
2 The terminology used to describe gene names and functions needs to be consistent throughout the manuscript. The switch between “bZIP55 and its homologous gene bZIP21” could be confusing. It’s crucial to clarify whether bZIP21 is a homolog, isoform, or distinct gene with similar functions to ensure clear communication.
Answer: Thanks for your professional comments. bZIP21 is a homolog of bZIP55. We have unified “bZIP55 and its homologous gene bZIP21” to “bZIP55 and bZIP21 (a homolog of bZIP55)” throughout the manuscript.
3 The description of bioinformatic approaches and the yeast two-hybrid assays could be insufficiently detailed for reproducibility. Ensuring that enough details are provided about the methods and tools used in bioinformatics analysis and laboratory experiments is essential for scientific validity.
Answer: Thanks for your professional comments. The description of bioinformatic approaches in the previous version “The gene structures of the PtoC/S1-bZIPs were analyzed using GSDS (https://gsds.gao-lab.org/index.php/, 20th, December, 2023). The conserved motifs were analyzed using MEME with default parameters (https://meme-suite.org/meme/index.html/, 20th, December, 2023).” have changed to “To analyze the gene structures of the PtoC/S1-bZIP subfamilies, the genomic sequences and corresponding coding sequences of PtoC/S1-bZIPs were downloaded from the genome database of P. tomentosa. Then the gene structure diagrams were drawn by the gene structure visualization server GSDS (https://gsds.gao-lab.org/index.php/, 20th, December, 2023). The protein sequences of PtoC/S1-bZIPs were submitted to MEME website (https://meme-suite.org/meme/index.html/, 20th, December, 2023) to predict the conserved motifs with default parameters (the maximum number of motifs set to ten).” in this revision on page 10 lines 382-389. In addition, the description of the yeast two-hybrid assays in the previous version “The plasmids of BD and AD recombinant constructs were co-transformed into the yeast strain AH109 and cultured on either SD/-Leu/-Trp solid medium or SD/-His/-Leu/-Trp/-Ade solid medium supplemented with X-α-gal.” has changed to “The plasmids of BD and AD recombinant constructs were co-transformed into the yeast strain AH109. The yeast cells were cultured on SD/-Leu/-Trp solid medium for four days, and positive yeast clones were verified by PCR. Then the positive yeast strains were grown in SD/-Leu/-Trp liquid medium and spotted on either SD/-Leu/-Trp solid medium or SD/-His/-Leu/-Trp/-Ade solid medium supplemented with X-α-gal. These plates with yeast cells were incubated at 28 °C for four days and then photographed.” in this revision on page 10 lines 405-411.
4 The term “artificial microRNA mediated gene silencing” is mentioned, but there's no detail on the specific constructs or the efficiency of the silencing, which are crucial for assessing the reliability of the experimental outcomes.
Answer: Thanks for your suggestion. In this study, the construction of amiRNA vector and the identification of PtobZIP55/21-amiRNA transgenic lines have been described in detail in the materials and methods. In the results, we have described that seven PtobZIP55/21-amiRNA transgenic lines were confirmed by RT-qPCR, and the results in figure S3 showed that the expression levels of PtobZIP55 and PtobZIP21 were both significantly decreased in amiRNA lines compared to WT plants. These results suggest that PtobZIP55 and PtobZIP21 were efficiently silenced by amiRNA technology.
5 “active the expression of seed maturation gene” in line 55 should be corrected to “activate the expression of seed maturation genes.”
Answer: Thanks for your careful reading. Now we have corrected “active” to “activate” in this revision on page 2 line 52.
6 The manuscript could benefit from consistent and clear definitions, especially when discussing complex mechanisms. For instance, the role of uORFs in regulating translation under different sucrose conditions could be expanded for clarity and depth.
Answer: Thanks for your professional comments. Now we have expanded the description of the role of uORFs in regulating translation under different sucrose conditions: “Studies have demonstrated that the luciferase activities are significantly reduced in sucrose-treated 35S::AtbZIP11 uORF::LUC transgenic lines compared to those without sucrose supplementation [17]. Moreover, biochemical analyses indicate that the AtbZIP11 uORF causes ribosome stalling in a sucrose-dependent manner, thus repressing the translation of the main AtbZIP11 ORF [18]. In contrast, under low carbon conditions, the translation of AtbZIP11 derepressed, coinciding with a decrease in sucrose content [19].” on page 2 lines 62-68.
7 In line 57, “This uORF encodes a sucrose control peptide,” could specify that it's not just “a” but specifically the uORFs discussed that encode this peptide, enhancing clarity.
Answer: Thanks for your constructive suggestion. To clarify the point, we have changed the sentence in the previous version “This uORF encodes a sucrose control peptide, which can inhibit the translation of the mORF under high sucrose condition.” to “These uORFs encode one or more oligopeptides, which can inhibit the translation of S1-bZIP main ORFs under conditions of high cytoplasmic sucrose levels” in this revision on page 2 lines 60-62.
8 The use of terms like “homodimers or heterodimers” in lines 42-43 should be explained briefly to ensure clarity for readers not familiar with the terminology.
Answer: Thanks for your thoughtful suggestion. To help readers better understand, we have changed the sentence in the previous version “The leucine zipper region typically forms an amphiphilic α-helix structure that allows the formation of homodimers or heterodimers between bZIP monomers,” to “The leucine zipper region typically forms an amphiphilic α-helix structure facilitating the formation of either homodimers between two identical bZIP monomers or heterodimers between two distinct bZIP monomers.” in this revision on page 1 lines 40-42.
9 References are too old in the introduction and discussion section, In so doing, it is suggested that the following articles be used as a reference.1. Pyrosequencing uncovers a shift in bacterial communities across life stages of Octodonta nipae (Coleoptera: Chrysomelidae).” 2. The endosymbiotic Wolbachia and host COI gene enable to distinguish between two invasive palm pests; coconut leaf beetle, Brontispa longissima and hispid leaf beetle, Octodonta nipae. Journal of Economic Entomology. Furthermore, the authors should try to provide some different keywords. This would increase the visibility of the paper by search engines if accepted for publication by the journal.
Answer: Thanks for your suggestion. Now we have updated and added relevant references based on your suggestion and cited them appropriately on page 9 lines 330-332. In addition, to increase the visibility of the paper by search engines if accepted for publication by the journal, we have changed the keywords “low energy” to “energy deprivation” as your thoughtful suggestion.
10 Conclusion section looks like the introduction, it must concise and only provide conclusion of study results and give some future direction, how your study can provide basic data for researcher working on microbial role.
Answer: Thanks for your professional suggestion. Now we have changed the sentence in the conclusion of previous version “In the present study, four C-bZIPs and ten S1-bZIPs were identified in P. tomentosa. The characteristics of C/S1-bZIP subfamily members, including phylogenetic relationships, multiple sequence alignments, gene structures, conserved motifs, and uORF identification, were analyzed using bioinformatic approaches. Additionally, Y2H assays revealed that each C-bZIP member can interact with S1-bZIP members. The expression profile analysis of C/S1-bZIPs showed that most members of C/S1-bZIP subfamilies were highly expressed in the roots and their expression was induced by low energy. Notably, bZIP55 and its homologous gene bZIP21 exhibited the highest expression levels among C/S1-bZIP members under low energy conditions. Further genetic analyses revealed a critical role of bZIP55/21 in responding to the low energy by inhibiting root growth and development. These results provide a foundation for further investigation of the biological functions of C/S1-bZIP genes in poplars and their molecular mechanisms in response to low energy.” to “In the present study, four C-bZIPs and ten S1-bZIPs were identified in P. tomentosa. Bioinformatic analyses indicated that PtoC/S1-bZIP subfamilies were conserved during the evolution. Additionally, the specific uORF sequences were identified in the 5’UTR region of some PtoS1-bZIPs members. The results of Y2H assays revealed that each C-bZIP member interacted with S1-bZIP members. Notably, most C/S1-bZIPs exhibited increased expression levels under low energy conditions, with bZIP55 and its homolog bZIP21 showing significant upregulation. Further genetic analyses suggested that bZIP55/21 play a role in inhibiting root growth and development in response to low energy. These findings provide a foundation for further investigation of the biological functions and molecular mechanisms of C/S1-bZIPs in response to environmental stress in poplars.” in this revision on page 11 and page 12 lines 455-464.
11 How do bZIP55 and bZIP21 specifically contribute to the regulation of root growth under low energy conditions? What are the downstream targets or pathways mediated by these genes during stress response?
Answer: Dear reviewer, we have added the discussion of your questions on page 9 lines 337-348, “The expression levels of close homologs AtbZIP2, AtbZIP11, and AtbZIP44 were suppressed by sucrose and induced by low energy, highlighting their roles in energy responses [17]. Recent studies have shown that the primary root growth under starvation is significantly less reduced in the AtbZIP2/11/44-amiRNA lines compared to WT [43]. Furthermore, molecular analyzes suggested AtbZIP2/11/44 inhibit root growth by activating the transcriptional repressor INDOLE-3-ACETIC ACID PROTEIN 3 (AtIAA3) under low energy conditions [43]. In P. tomentosa, the paralogous genes PtobZIP55 and PtobZIP21, sharing close homology with AtbZIP2/11/44, were the most up-regulated C/S1-bZIP genes under low energy conditions, suggesting similar biological functions to those of AtbZIP2/11/44. Silencing of PtobZIP55/21 by amiRNA technology increased the AR length, LR length and LR number in the transgenic poplars under low energy condition, further indicating their negative impacts on root growth and development.”
12 What are the molecular mechanisms underlying the interactions between C-bZIPs and S1-bZIPs? How do these interactions influence the transcriptional regulation under low energy conditions?
Answer: Dear reviewer, it has been reported that C-bZIPs can form heterodimers with S1-bZIPs to promote metabolic reprogramming to maintain energy homeostasis in response to low energy in Arabidopsis thaliana and Malus domestica. Consistently, our results indicate that each C-bZIP member interacts with S1-bZIPs in Populus tomentosa. Under extended dark conditions, the expression levels of most PtoC/S1-bZIPs are up-regulated. It can be inferred that in the nucleus, C-bZIPs interact with S1-bZIPs (such as PtobZIP55) through the leucine zipper domain to form heterodimers that subsequently activate the expression of downstream target genes involved in catabolic metabolism. Now we have added these points in the discussion on page 8 lines 300-310 and page 9 lines 311-329, respectively.
13 Is there functional redundancy among the C/S1-bZIP members, and if so, how does this redundancy contribute to the robustness of the low energy stress response in Populus tomentosa?
Answer: Dear reviewer, previous studies have been reported that C/S1-bZIP heterodimerization network acts as a signaling hub for coordinating plant development and stress response in a functionally redundant manner in Arabidopsis thaliana and Malus domestica. In our study, phylogenetic analysis revealed that members within the C and S1 subfamilies exhibit high sequence similarity and close relationships. Furthermore, most members of PtoC/S1-bZIP subfamilies showed partially overlapping expression patterns. Therefore, we speculate that PtoC/S1-bZIP members may function redundantly. Now we have added these points in the discussion on page 9 lines 311-329.
14 How do the functions of C/S1-bZIP members in Populus tomentosa compare with those in other plant species, particularly in terms of evolutionary adaptations to energy stress?
Answer: Dear reviewer, C/S1-bZIPs in A. thaliana are involved in regulating carbohydrate and amino acid metabolism by directly bind to the protomers of PROLINE DEHYDROGENASE (ProDH) and ASPARAGINE SYNTHETASE1 (ASN1) under energy limitation. Similar results are also found in in Solanum lycopersicum and Nicotiana tabacum. It also has been demonstrated that AtbZIP2/11/44 are induced by low energy and inhibit root growth under energy deprivation. Consistently, most PtoC/S1-bZIP members are up-regulated under extended dark conditions, and the results of PtobZIP55/21-amiRNA transgenic analyses suggest that bZIP55/21 negatively regulate root development of P. tomentosa in response to low energy. Now we have added these points in the discussion on page 9 lines 332-348.
15 Beyond root growth inhibition, how does the suppression of bZIP55 and bZIP21 affect other physiological and developmental processes in Populus tomentosa under low energy conditions?
Answer: Dear reviewer, except for the root phenotypes, no differences have been found in the shoots of PtobZIP55/21 transgenic lines compared to WT poplars so far. In the future, we consider removing the poplar seedlings from WPM medium and cultivating them in soil under low energy treatment for the long term to further observe phenotypic changes in different tissues of PtobZIP55/21 transgenics and WT poplars.
16 What are the upstream signals and factors that regulate the expression of bZIP55 and bZIP21 during low energy stress?
Answer: Dear reviewer, now we have added the discussion on page 9 lines 315-317, “Snf1-RELATED PROTEIN KINASE 1 (SnRK1), a central metabolic kinase, plays a pivotal role in responding to low energy signal. Transcriptomic and molecular analyses have identified S1-bZIPs as SnRK1-dependent regulators under energy deprivation” In our study, PtobZIP55 and PtobZIP21 share close homology with AtbZIP2/11/44 of AtS1-bZIPs. Low energy induced transcription levels of PtobZIP55 and PtobZIP21, which are consistent with the expression profiles of AtbZIP2/11/44. Moreover, PtobZIP55/21 show similar biological function of inhibiting root growth by genetic analysis. Thus, we speculate that SnRK1 is probably upstream mediator of PtobZIP55 and PtobZIP21 in response to low energy in P. tomentosa.

Reviewer 3 Report
Comments and Suggestions for Authors
Abstract
The abstract section provides the general overview of the study. This study examines a specific category of plant proteins known as bZIP proteins to investigate their reactions to low energy stress. The proteins of four C-bZIPs and ten S1-bZIPs were examined to understand their relationships, gene structures, patterns, and regulatory DNA sequences Additionally, it has been identified that the yeast studies demonstrated that C-bZIP and S1-bZIP cooperate and have a mutual interaction. The study found that most of the C/S1-bZIPs were not transcribed when energy levels were low, especially bZIP55 and its similar bZIP21.
Introduction
The introduction section provides the introductory part of the study. It can be depicted that the bZIP family is crucial for regulating numerous cellular processes. It includes proteins called transcription factors which have a special shape and help cells respond to different situations. This design typically produces a structure known as an alpha-helix, which facilitates the connection of two molecule components and enables the molecule to perform various functions. This study explores the role of C and S1-bZIP proteins in Populus tomentosa trees, specifically their function in efficient energy utilization.
Result
The result section describes the outcomes of the conducted research. The research is examining and explaining C/S1-bZIP genes in P.A is used to tomentosa. It has been identified that Thaliana DNA sequences are used as lures. Using tools like Expasy and MEGA11 helps to understand how proteins work and if they are stable in different cell conditions. Analyzing the structures of genes and proteins can provide insight into their collaborative role in regulating various bodily functions. The study also forecasts the structure of S1-bZIP proteins and how they interact with other genes, providing insight into their functioning.
Discussion
The discussion chapter provides the knowledge of the result and analyse the result accordingly. It has been identified that the study investigates a specific group of genes in the Populus tomentosa plant and their role in regulating root growth under conditions of limited energy.A total of four C-bZIPs and ten S1-bZIPs were detected, with a lengthy presence and unique interactions, especially when combined in pairs. Switching off specific bZIP genes resulted in increased root growth, indicating that they typically inhibit root growth during times of low energy. However, it is essential for plants to be capable of coping with harsh environmental conditions.
Materials and Method
The materials and method section provides the significant methodology which has been used to conduct the research. It has been identified that researchers utilised the BLASTP method to search for C/S1-bZIP subfamilies in Populus tomentosa. The proteins underwent examination with HMMER, PFAM, and SMART tools, while ExPASy was employed to anticipate their characteristics. Additionally, examining the differences among different types of animals, including A Thaliana and others were studied using MEGA11 to look at their family tree, and ClustalX was used to compare their genetic sequences.
Comments
The document provides a thorough analysis of certain protein groups in Populus tomentosa, employing a robust technique to categorize and delineate these proteins. Bioinformatics can be strengthened with the addition of experimental data. However, the significance of these proteins in other aspects of plant biology, such as their role in assisting trees to manage stress and promote growth need to be clarified.
Comments on the Quality of English LanguageMinor editing of English language required
Author Response
Abstract
The abstract section provides the general overview of the study. This study examines a specific category of plant proteins known as bZIP proteins to investigate their reactions to low energy stress. The proteins of four C-bZIPs and ten S1-bZIPs were examined to understand their relationships, gene structures, patterns, and regulatory DNA sequences Additionally, it has been identified that the yeast studies demonstrated that C-bZIP and S1-bZIP cooperate and have a mutual interaction. The study found that most of the C/S1-bZIPs were not transcribed when energy levels were low, especially bZIP55 and its similar bZIP21.
Introduction
The introduction section provides the introductory part of the study. It can be depicted that the bZIP family is crucial for regulating numerous cellular processes. It includes proteins called transcription factors which have a special shape and help cells respond to different situations. This design typically produces a structure known as an alpha-helix, which facilitates the connection of two molecule components and enables the molecule to perform various functions. This study explores the role of C and S1-bZIP proteins in Populus tomentosa trees, specifically their function in efficient energy utilization.
Result
The result section describes the outcomes of the conducted research. The research is examining and explaining C/S1-bZIP genes in P.A is used to tomentosa. It has been identified that Thaliana DNA sequences are used as lures. Using tools like Expasy and MEGA11 helps to understand how proteins work and if they are stable in different cell conditions. Analyzing the structures of genes and proteins can provide insight into their collaborative role in regulating various bodily functions. The study also forecasts the structure of S1-bZIP proteins and how they interact with other genes, providing insight into their functioning.
Discussion
The discussion chapter provides the knowledge of the result and analyse the result accordingly. It has been identified that the study investigates a specific group of genes in the Populus tomentosa plant and their role in regulating root growth under conditions of limited energy. A total of four C-bZIPs and ten S1-bZIPs were detected, with a lengthy presence and unique interactions, especially when combined in pairs. Switching off specific bZIP genes resulted in increased root growth, indicating that they typically inhibit root growth during times of low energy. However, it is essential for plants to be capable of coping with harsh environmental conditions.
Materials and Method
The materials and method section provides the significant methodology which has been used to conduct the research. It has been identified that researchers utilised the BLASTP method to search for C/S1-bZIP subfamilies in Populus tomentosa. The proteins underwent examination with HMMER, PFAM, and SMART tools, while ExPASy was employed to anticipate their characteristics. Additionally, examining the differences among different types of animals, including A Thaliana and others were studied using MEGA11 to look at their family tree, and ClustalX was used to compare their genetic sequences.
General answer: Thank you for spending your precious time on reviewing our manuscript. We greatly appreciate your positive comments. Maintaining energy homeostasis is crucial for plant survival. It has been reported that C/S1-bZIPs play important roles in managing the low energy response in Arabidopsis thaliana. However, the comprehensive characterization of C/S1-bZIP members in woody plants and their functions in response to low energy have yet to be defined. In this study, we identified the members of C/S1-bZIP subfamilies using bioinformatic approaches in Populus tomentosa, and characterized their interactions by Y2H assays. We also investigated the biological functions of key genes bZIP55 and bZIP21 in response to low energy by genetic analyses. Based on the above findings, we want to provide insights into the molecular mechanisms by which C/S1-bZIPs regulate poplar growth and development in response to energy deprivation.
Comments
The document provides a thorough analysis of certain protein groups in Populus tomentosa, employing a robust technique to categorize and delineate these proteins. Bioinformatics can be strengthened with the addition of experimental data. However, the significance of these proteins in other aspects of plant biology, such as their role in assisting trees to manage stress and promote growth need to be clarified.
Answer: Thanks for your professional comments. Now we have strengthened the description of bioinformatic approaches in this revision on page 10 lines 382-389. In addition, The C/S1-bZIP network has been established as a signaling hub for coordinating plant development and stress response. In Arabidopsis thaliana, Transcriptomic and molecular analyses have identified AtS1-bZIPs as AtSnRK1-dependent regulators under energy deprivation. AtbZIP1 and AtbZIP53 reprogram carbohydrate and amino acid metabolism to serve the energy demand under salt stress. In M. domestica, the MdC/S1-bZIP network has been found to negatively regulate drought tolerance and low energy-induced senescence in a functionally redundant manner. In our studies, genetic analyses revealed that bZIP55/21 inhibited root growth and development under low energy stress. Thus, we speculate that PtoC/S1-bZIPs probably play crucial roles in controlling energy homeostasis in response to fluctuating environmental conditions, thereby optimizing the balance between plant growth and stress tolerance. Now we have added these points in the discussion on page 9 lines 315-329 and lines 339-348, respectively.

Round 2
Reviewer 2 Report
Comments and Suggestions for Authors
i am satisfied the revision completed by authors